# Investigation into the Suppression Effects of Inert Powders on the Minimum Ignition Temperature and the Minimum Ignition Energy of Polyethylene Dust

**Chendi Lin [1,2], Yingquan Qi [1,2], Xiangyang Gan [1,2], Hao Feng [1,2], Yan Wang [1,2,*], Wentao Ji [1,2,*] and Xiaoping Wen [3]**

1   State Key Laboratory Cultivation Bases for Gas Geology and Gas Control, College of Safety Science and Engineering, Henan Polytechnic University, Jiaozuo 454000, China; lin_chendi@163.com (C.L.); nuoweihaidao@hotmail.com (Y.Q.); gxyoung@aliyun.com (X.G.); fenghaohpu@163.com (H.F.)

2   The Collaboration Innovation Center of Coal Safety Production of Henan Province, Henan Polytechnic University, Jiaozuo 454000, China

3   School of Mechanical and Power Engineering, Henan Polytechnic University, Jiaozuo 454003, China; wenxiaoping666@163.com

*   Correspondence: yanwang@hpu.edu.cn (Y.W.); jiwentao@hpu.edu.cn (W.J.)

**Abstract:** The risks associated with dust explosions still exist in industries that either process or handle combustible dust. This explosion risk could be prevented or mitigated by applying the principle of inherent safety. One effective principle is to add an inert material to a highly combustible material in order to decrease its ignition sensitivity. This paper deals with an experimental investigation of the influence of inert dust on the minimum ignition temperature and the minimum explosion energy of combustible dust. The experiments detailed here were performed in a Godbert–Greenwald (GG) furnace and a 1.2 L Hartmann tube. The combustible dust (polyethylene—PE; 800 mesh) and four inert powders ($NaHCO_3$, $Na_2C_2O_4$, $KHCO_3$, and $K_2C_2O_4$) were used. The suppression effects of the four inert powders on the minimum ignition temperature and the minimum explosion energy of the PE dust have been evaluated and compared with each other. The results show that all of the four different inert dusts have an effective suppression effect on the minimum ignition temperature and the minimum explosion energy of PE dust. However, the comparison of the results indicates that the suppression effect of bicarbonate dusts is better than that of oxalate dust. For the same kind of bicarbonate dusts, the suppression effects of potassium salt dusts are better than those of the sodium salt. The possible mechanisms for the better suppression effects of bicarbonate dusts and potassium salt dusts have been analyzed here.

**Keywords:** dust explosion; minimum ignition temperature; minimum ignition energy; inert powder; suppression effect

## 1. Introduction

Polyethylene (PE) is a type of thermoplastic resin made from polymerized ethylene. It is one of the top five major synthetic resins in the world. During the process of producing PE particles, high concentrations of dust clouds can be formed in the process of granulation, drying, pneumatic transportation, and unloading. This dust can be easily ignited by ignition sources such as static electricity or hot mechanical surfaces [1]. This causes dust explosion accidents and serious casualties and property loss. A good way to prevent these accidents and mitigate the associated risks is to reduce the ignition sensitivity of combustible dusts. The minimum ignition temperature (MIT) and the minimum ignition energy (MIE) are the key parameters which reflect the ignition sensitivity of

combustible dusts. Therefore, taking some measures to raise the MIT and MIE of combustible dusts is necessary to reduce the ignition sensitivity of combustible dusts.

In the past few decades, many investigations have focused on increasing the MIT and MIE of combustible dusts by adding an inert substance. Yang et al. [2,3] investigated the inerting effects of $CO_2$ on the MIT and MIE of micron-size acrylate copolymer (ACR) dusts. When the $CO_2$ concentration reached 40% of the volume, the MIT and MIE of ACR dusts increased by 50 °C and 257 mJ, respectively. Meanwhile, they investigated the inerting effects of ammonium polyphosphate (APP) on the MIT and MIE of polypropylene (PP) dusts. The MIT and MIE of PP dusts increased when the powder concentration of APP increased. Addai et al. [4] investigated the effects of adding inert powders (magnesium oxide, ammonium sulphate, and sand) on the ignition sensitivity (MIT and MIE) of combustible dust (brown coal, lycopodium, toner, niacin, corn starch, and high density polyethylene) mixtures. The results showed that when the concentration of inert powders was between 60 to 80%, the ignition risk could be reduced to a minimum. Yu et al. [5] studied the inerting effect of crystalline II type ammonium polyphosphate (APP-II) on the MIE of micron-sized acrylate copolymer (ACR) dust. The results showed that the inert powder had a significant inerting effect on the MIE of ACR dust, and when the APP-II concentration reached 40%, the minimum MIE value of 960 mg for ACR dust increased to 990 mJ. Yuan et al. [6] investigated the inerting effect of nano-sized $TiO_2$ powder on the MIE of nano- and micro-sized Ti dusts, and the results showed that nano-sized $TiO_2$ powder could effectively reduce the ignition sensitivity of micro-sized Ti dust to electric sparks, but nano-sized $TiO_2$ powder might not be suitable for nano-sized Ti particles, because the mixtures still have high sensitivity, even with 90% nano-sized $TiO_2$ powder. Yu et al. [7] measured the MIT of coal dust clouds containing various quantities of coal fly ash, and coal fly ash was found to have a better inhibiting effect than calcium carbonate. Liu et al. [8] investigated the inerting effect of inert powders ($NH_4H_2PO_4$, NaCl and $CaCO_3$) on the MIT of coal and oil shale mixed dust clouds (COSMD), and the critical concentrations of inert powders were 30%, 40%, and 60%, respectively. Bu et al. [9] investigated the inhibitory effect of $Al_2O_3$ at four particle size distributions on the minimum ignition energy (MIE) of aluminum dust and then analyzed the effect of particle size on the inerting efficiency. The results indicated that nano-sized $Al_2O_3$ powder has a superior inerting efficacy compared to that of micro-sized $Al_2O_3$.

According to the existing research, it can be found that the addition of inert powders can effectively decrease the ignition sensitivity of combustible dusts. Hence, in this study, the effects of inert powders on the MIT and MIE of PE dusts are investigated. Four different kinds of powders have been selected for this study. Among these, sodium bicarbonate ($NaHCO_3$) and potassium bicarbonate ($KHCO_3$) were selected because they are commonly used as BC powders, and they both have presented a significant suppression effect on gas and dust explosions throughout many research studies [10–15]. In addition, two different oxalate powders ($Na_2C_2O_4$ and $K_2C_2O_4$) were also selected because of their suppression effect on flame propagation [16–18]. According to the experimental results and by comparing the inhibitory effects of four inert powders, the suppression effects and mechanisms of alkali metal ions (sodium ion and potassium ion) and acid radical ions on the MIT and MIE of PE dusts (bicarbonate ion and oxalate ion) are analyzed here.

## 2. Materials and Experimental Work

### 2.1. Measurement of the Minimum Ignition Temperature

In this study, a modified Godbert–Greenwald (GG) furnace was used. It was used in accordance with EN 50281 [19], as shown in Figure 1. It consisted of a GG furnace and a gas supply system. The GG furnace consisted of a cylindrical furnace with an external stainless-steel structure. The internal wall was made of refractory ceramic, which could sustain temperatures up to 1000 °C. A set of thermocouples were installed in the furnace to monitor the internal temperature. The furnace was vertically positioned, and its bottom was open. The upper end was connected to a dust chamber via a glass adaptor. The dust was dispersed into the furnace by a high-pressure air pulse, which was

obtained by opening a solenoid valve to discharge the air stored in a reservoir. A metal mirror plate was placed under the furnace to allow the operator to observe the inside of the furnace and to judge whether the dust mixture was ignited.

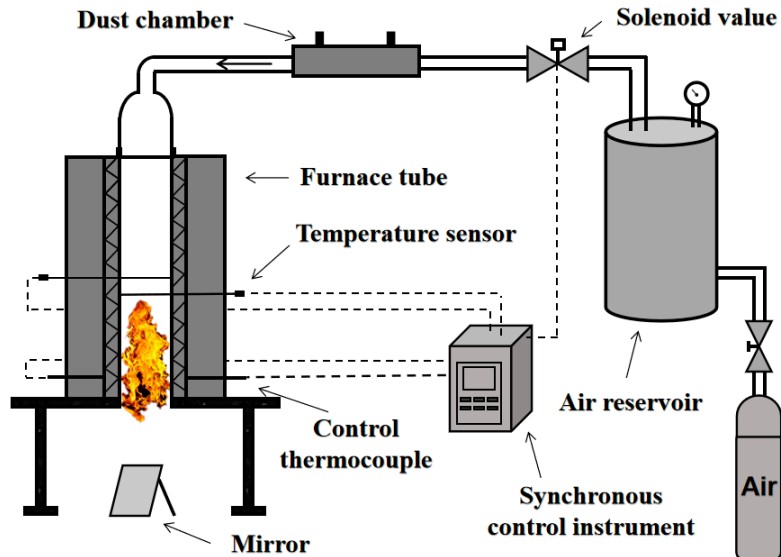

**Figure 1.** Experimental apparatus for the minimum ignition temperature (MIT) test.

In order to ensure the repeatability and reproducibility of the experiments, all experiments were carried out under the same initial conditions according to the following steps.

First, the furnace tube was heated to the desired temperature and an amount of dust, weighed beforehand, was placed in the dust chamber. Second, the air reservoir was filled with air up to the desired dispersion pressure, and the dust sample was then dispersed through the furnace tube by the blast of air. The criterion for indicating an explosion was the observation of a flame at the bottom open mouth of the furnace with the help of a mirror. In addition, both the pressure in the air reservoir (0.1–0.5 bar above atmospheric pressure) and the mass of dust (0.1–0.5 g) were varied until a vigorous explosion was obtained. The condition at which the vigorous explosion occurred was taken as the "best" explosion condition. This condition was maintained, except the furnace temperature was lowered and testing was continued until no flame was observed after ten successive attempts. As soon as the MIT was obtained, further tests were performed at a furnace temperature 10 °C below the MIT by varying both the pressure and the mass of dust mixtures to confirm the non-ignition state.

### 2.2. Measurement of the Minimum Ignition Energy

The minimum ignition energy was determined with an electric spark igniter (Hartmann apparatus, Institute of Industrial Explosion and Protection, Northeastern University, Shenyang, China) according to a protocol similar to that defined in the EN 13821 standard [20]. The schematic diagram of the Hartmann apparatus is shown in Figure 2. In the Hartmann apparatus, the combustion chamber was a glass tube with a volume of 1.2 L, and it was provided with a mushroom-shaped dust dispersion system. Dust dispersion was triggered by a 7-bar compressed air blast. The air blast generated considerable turbulence and resulted in the creation of a dust cloud. A spark was drawn between two electrodes. According to the standard, the spark gap was set at 6 mm. All tests were performed at atmospheric pressure and room temperature and by following the method defined in the EN 13821 standard.

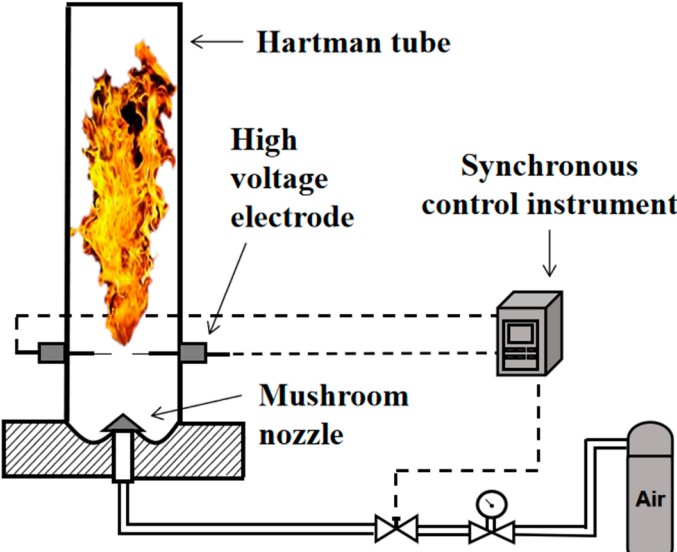

**Figure 2.** Experimental apparatus for the minimum ignition energy (MIE) test.

The minimum ignition energy (MIE) is between the lowest energy value (E2) at which ignition occurred and the energy (E1) at which no ignition was observed in at least 10 successive experiments. The ratio of energy steps should be ≤3.3, for instance, 1 mJ, 3 mJ, 10 mJ, and so on. The MIE was usually stated as a range of values rather than as a single value. However, in order to compare the MIE of the different combustible powders more clearly, only a single value could be used. In the EN 13821 standard, this value can be estimated using the probability of ignition, as stated below:

$$MIE = 10^{\log E2 - \frac{I[E2] \times (\log E2 - \log E1)}{(NI+I)[E2]+1}}$$

(1)

where *E2* is the energy level at which ignition occurs and *E1* is the energy level below *E2* with no ignition after 10 consecutive attempts. *I[E2]* represents the number of tests with ignition at *E2* and *(NI + I)[E2]* represents the total number of tests at *E2*. For the MIE calculations, *(NI + I)[E2]* should be greater than or equal to 5 tests.

### 2.3. Materials

Four inert materials were used in this study, namely, sodium bicarbonate ($NaHCO_3$), sodium oxalate ($Na_2C_2O_4$), potassium bicarbonate ($KHCO_3$), and potassium oxalate ($K_2C_2O_4$) (Kermel Chemical Reagent Co., Ltd., Tianjin, China). The particle sizes of these inert powders and of the PE dust are shown in Figure 3. They were measured using the Mastersizer 2000 instrument (Malvern Panalytical Ltd., Almelo, The Netherlands). According to Figure 3, it can be determined that the median diameter of these particles was 77.5 μm for $NaHCO_3$, 77 μm for $Na_2C_2O_4$, 75.1 μm for $KHCO_3$, and 73.5 μm for $K_2C_2O_4$, respectively. Hence, the four inert powders can be regarded as having the same particle size because there is little difference in their median diameters. The median diameters of the 80 mesh and 400 mesh $KHCO_3$ powders were 178 μm and 38.8 μm, respectively. The median diameter of PE dust was 16.7 μm. Figure 4 shows the structure of these particles via SEM observation, and the results were obtained using field-emission scanning electron microscopy (SEM, JEOL, JSM-6390LV, Tokyo, Japan).

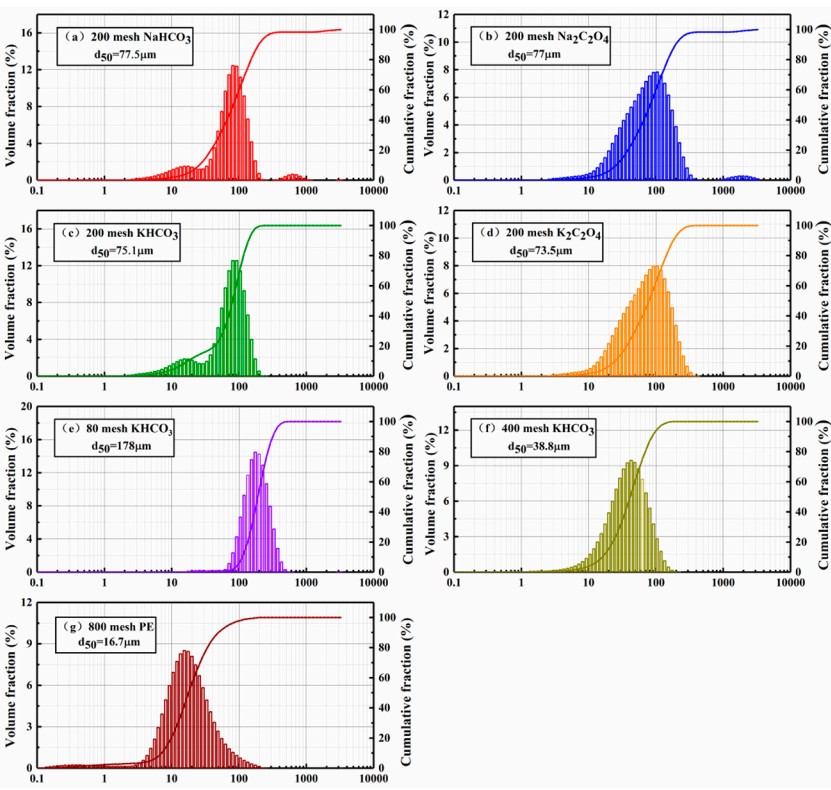

**Figure 3.** Particle size distributions of different dust particles.

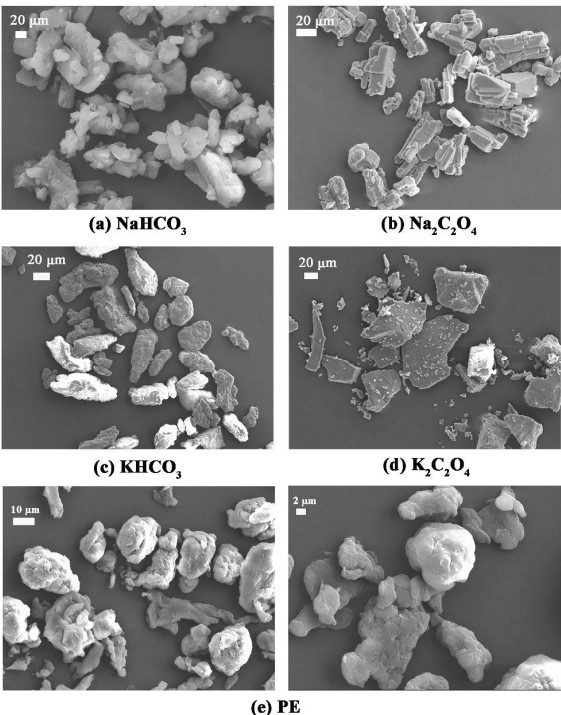

**Figure 4.** SEM images of different dust particles.

All of the dust samples were dried at the temperature of 50 °C for 24 h before the experiments. For every mixture combination, both the PE dusts and the inert powders were weighed separately and then placed in a transparent glass bottle. The bottle was shaken vigorously in all possible directions to achieve homogenous mixing.

## 3. Results and Discussion

### 3.1. Effect of Inert Powders on the MIT and MIE of PE Dusts

Figure 5a presents the effects of the four inert powders on the MIT of PE dust. It can be seen that the MIT of PE dust obviously increases after the addition of the four different inert powders and that it increases further as the inert powder concentration increases. This indicates that all of the four different inert powders have a suppression effect on the MIT of the PE dust. Figure 5b presents the effects of the four inert powders on the MIE of PE dust. It can also be observed that the MIE of PE dust increases with an increase in the inert powder concentration, indicating that all of the four different inert powders also have a suppression effect on the MIE of PE dust.

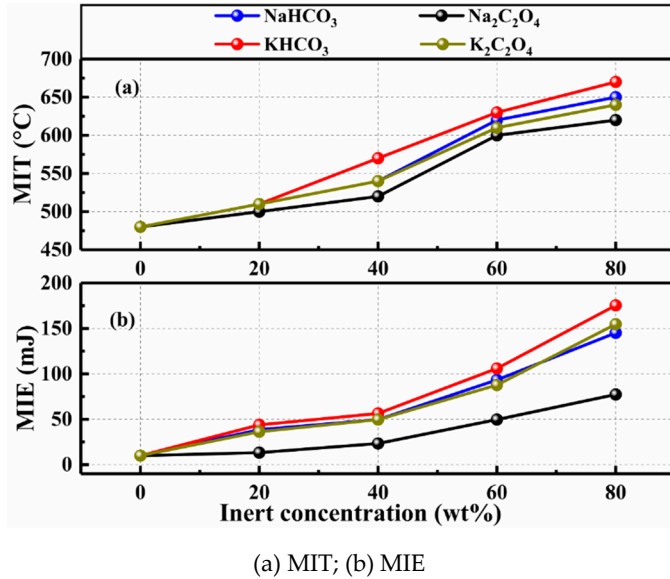

(a) MIT; (b) MIE

**Figure 5.** Effect of the concentration of four inert powders on the MIT and MIE of polyethylene (PE).

According to the change curves of the MIT and MIE of PE dusts, it can be observed that all of the inert powders have good suppression effects, but they present different rising tendencies with an increase in inert powder concentrations. This could be ascribed to the effects of alkali metal ions (sodium ion and potassium ion) and acid radical ions (bicarbonate ion and oxalate ion), which will be analyzed in later chapters.

As we know, the particle size of inert powders is an important parameter which can greatly impact the suppression effect of inert powders on the MIT and MIE. In order to clarify the role of the particle sizes of the inert powders in the suppression effect, the effect of $KHCO_3$ powders with three particles sizes (80 mesh, 200 mesh, 400 mesh) on the MIT and MIE of PE dusts were sequentially investigated, as shown in Figure 6. $KHCO_3$ was chosen here because the suppression effect of $KHCO_3$ powder is better than that of the other three materials, therefore, it is beneficial to study it in order to detect the role of the particle size of inert powders in the suppression effect.

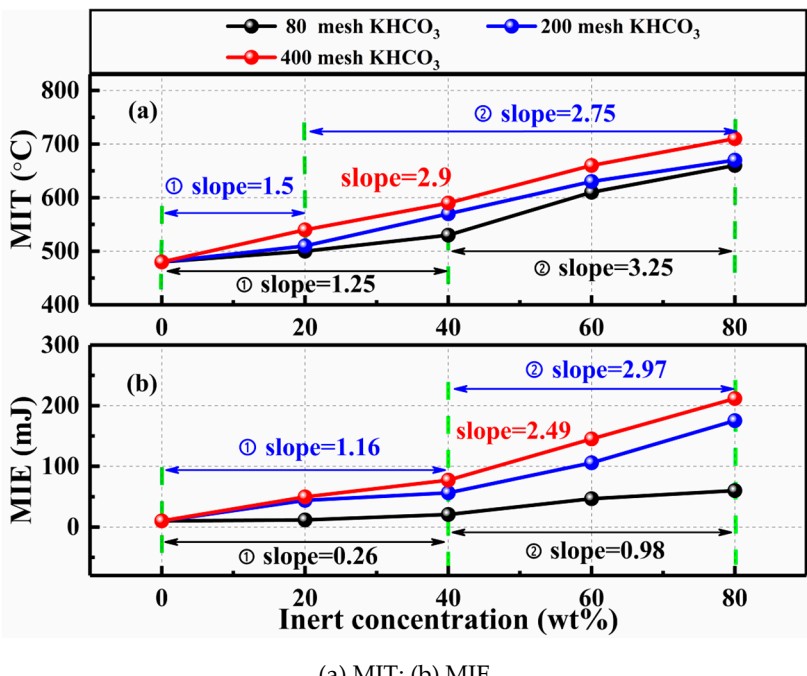

(a) MIT; (b) MIE

**Figure 6.** Effect of KHCO$_3$ powder of three particle sizes on the MIT and MIE of PE.

Figure 6 shows the effect of three particle sizes of KHCO$_3$ powder on the MIT and MIE of PE dust. It can be observed from Figure 6a that when the KHCO$_3$ powder concentration of all particle sizes rises, the MIT values of the PE dusts increase. According to the slopes of the 80 mesh KHCO$_3$ powder's linearly fitted lines, it can be found that their corresponding slopes are 1.25 and 3.25, respectively, indicating that when the concentration of KHCO$_3$ powder reaches 40%, the MIT values rise rapidly, where the concentration of the inert dust at this point is called the critical concentration. In other words, when the inert powder reaches its critical concentration, the inert powder exhibits a higher suppression effect, and it can effectively suppress dust ignition and combustion [8]. The critical concentration of 200 mesh KHCO$_3$ powder is lower than that of 80 mesh KHCO$_3$ powder, and the values are 20%. There is only one slope in the linear fit line of 400 mesh KHCO$_3$ powder, indicating that this particle size has a stronger suppression effect at the initial concentration. When all of the concentrations of three particle size powders (80 mesh, 200 mesh, 400 mesh) are 80%, the MIT values of PE dusts are 660 °C, 670 °C, and 710 °C, respectively. According to the increasing tendencies of the three powders on the MIT of PE, it can be discovered that there is little difference between the inhibitory effects of 200 mesh KHCO$_3$ powder and 80 mesh KHCO$_3$ powder, but the 400 mesh KHCO$_3$ powder has a greater suppression effect than the other two powders. In addition, it can be observed in Figure 6b that when the KHCO$_3$ powder concentration of all particle sizes rises, the MIE values of PE dusts increase. According to the slopes of their linear fitted lines, it can be found that the critical concentration of 200 mesh KHCO$_3$ powder is 40%, which is similar to that of 80 mesh KHCO$_3$ powder, but the slopes values of linear fitted line 1 are 1.16 and 0.26, respectively. This indicates that at lower concentrations, 200 mesh KHCO$_3$ powder has a stronger inhibitory effect than that of 80 mesh KHCO$_3$ powder. For 400 mesh KHCO$_3$ powder, there is only one slope in the linearly fitted line, indicating that the MIE of PE would rise rapidly, presenting a stronger suppression effect than the other two powders. When all of the concentrations of the three particles size powders (80 mesh, 200 mesh, and 400 mesh) reach 80%, the MIE values of PE increase to 60 mJ, 175 mJ, and 211 mJ, respectively. According to the increasing tendencies of three particle size powders on the MIE of PE, it can be discovered that the particle size of KHCO$_3$ powder has a great influence on the MIE of PE. The 400 mesh KHCO$_3$ powder shows a greater inhibitory effect than other two powders. To sum up, the KHCO$_3$ powder, with a smaller particle

size distribution, has a better inhibitory effect on the MIE and MIT of PE dust. This is similar to other author's conclusions [21–23].

### 3.2. Comparison between the Two Acid Radical Ions

In order to compare and analyze the effects of two acid radical ions (bicarbonate ion and oxalate ion) on the MIT and MIE of PE dusts, the four inert powders were divided into two groups (NaHCO$_3$ and Na$_2$C$_2$O$_4$ as the first group and KHCO$_3$ and K$_2$C$_2$O$_4$ as the other). The comparison results are shown in Figures 7 and 8, respectively.

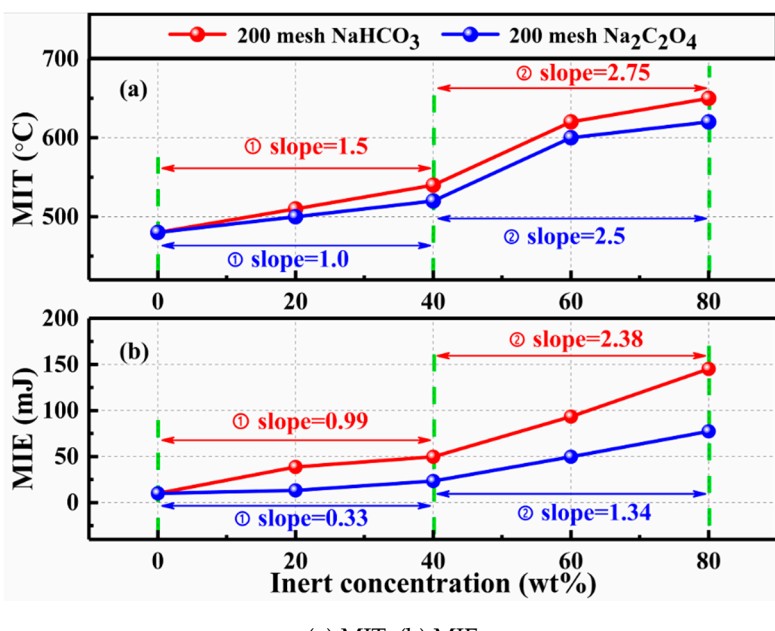

(a) MIT; (b) MIE

**Figure 7.** Effect of NaHCO$_3$ and Na$_2$C$_2$O$_4$ powder concentrations on the MIT and MIE of PE.

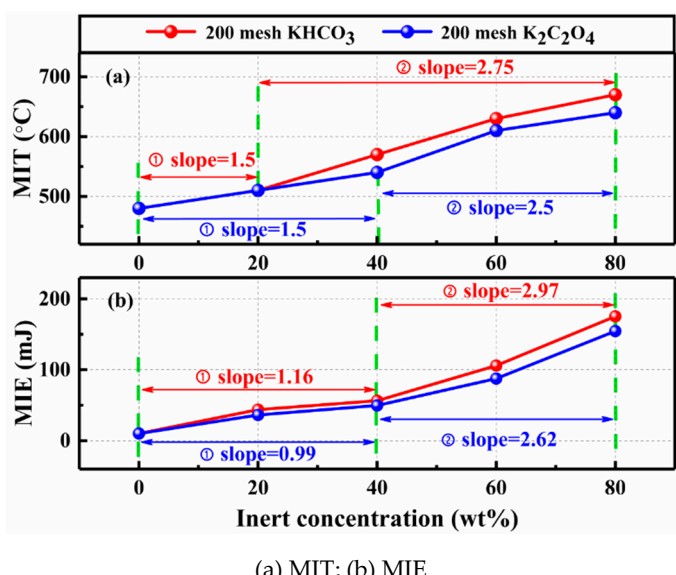

(a) MIT; (b) MIE

**Figure 8.** Effect of KHCO$_3$ and K$_2$C$_2$O$_4$ powder concentrations on the MIT and MIE of PE.

3.2.1. Effects of NaHCO$_3$ and Na$_2$C$_2$O$_4$ on the MIT and MIE of PE Dusts

Figure 7a shows the effect of NaHCO$_3$ and Na$_2$C$_2$O$_4$ powder concentrations on the MIT of PE dust. According to the slopes of the NaHCO$_3$ powder's linearly fitted lines, it can be found that their

corresponding slopes are 1.5 and 2.75, respectively. The corresponding critical concentration is 40%. At this concentration, the MIT of PE dust rises by 60 °C when compared with that at 0%. While, for the $Na_2C_2O_4$ powder, the corresponding slopes are 1.0 and 2.5, respectively, which are lower than that of the $NaHCO_3$ powder. The corresponding critical concentration of $Na_2C_2O_4$ is also 40%, which is same as that of the $NaHCO_3$ powder. Although the critical concentration of $Na_2C_2O_4$ is the same as that of $NaHCO_3$, the value of MIT rises by 40 °C at the critical concentration of $Na_2C_2O_4$ powder, which is lower than that of the $NaHCO_3$ powder. In addition, the MIT of PE dust reaches 650 °C when the $NaHCO_3$ concentration is 80%. While for $Na_2C_2O_4$, the MIT of PE dust only reaches 620 °C when the $Na_2C_2O_4$ concentration is 80%.

Figure 7b shows the effect of the $NaHCO_3$ and $Na_2C_2O_4$ powder concentration on the MIE of PE dust. According to linearly fitted lines 1 and 2, it can be observed that their corresponding slopes are 0.99 and 2.38, respectively, for $NaHCO_3$ powder. The critical concentration of $NaHCO_3$ powder is 40%. At this concentration, its MIE rises by 39 mJ when compared with the value at 0%, and the MIE of PE dust finally reaches 145 mJ when the $NaHCO_3$ powder concentration is 80%. For $Na_2C_2O_4$, according to the slopes of linearly fitted lines 1 and 2, their corresponding slopes are 0.33 and 1.34, respectively, which are lower than that of the $NaHCO_3$ powder. The critical concentration of $Na_2C_2O_4$ powder also is 40%. When the $Na_2C_2O_4$ powder concentration is 80%, the MIE of PE dust only reaches 77 mJ, which is lower than that of the $NaHCO_3$ powder.

By comparing the effects of two sodium salt powders, it can be seen that the inhibitory effect of $NaHCO_3$ on the MIT of PE dusts is better than that of $Na_2C_2O_4$. Since they both have the same metal cation, it could be speculated that the inhibitory effect of bicarbonate salt powders is better than that of oxalate salts powders when the inert powders have the same particle size distribution and the same metal cation. However, it is obvious that further studies are needed to test the suppression effect of another group of powders ($KHCO_3$ and $K_2C_2O_4$) and determine the veracity of the above conclusion.

### 3.2.2. Effects of $KHCO_3$ and $K_2C_2O_4$ on the MIT and MIE of PE Dusts

In order to verify the conclusion mentioned above, the effect of another group powders ($KHCO_3$ and $K_2C_2O_4$) of the same particle size distribution on the MIT and MIE of PE dusts were compared. The results are shown in Figure 8. In Figure 8a, with the increase of two potassium salt concentrations, the MIT of the PE dust cloud increases gradually, but there is a different rising tendency on the MIT of PE dusts. For $KHCO_3$, according to linearly fitted lines 1 and 2, the corresponding slopes are 1.5 and 2.75, respectively. The critical concentration of $KHCO_3$ is 20%, where its MIT rises by 30 °C when compared with that at 0%. When the $KHCO_3$ concentration increases to 80%, its MIT reaches 670 °C, which is a rise of 190 °C when compared with that at 0%. For $K_2C_2O_4$, according to linearly fitted lines 1 and 2, it can be found that the corresponding slopes are 1.5 and 2.5, respectively. The critical concentration of $K_2C_2O_4$ is 40%, which is higher than that of $KHCO_3$. When the $K_2C_2O_4$ powder concentrations are 60% and 80%, the MIT values are 610 °C and 640 °C, respectively.

Figure 8b presents the suppression effect of $KHCO_3$ and $K_2C_2O_4$ with the same particle size distribution on MIE of PE dusts. For $KHCO_3$, according to linearly fitted lines 1 and 2, it can be observed that the corresponding slopes are 1.16 and 2.97, respectively. The critical concentration of $KHCO_3$ is 40%, and, at this point, the MIE rises by 46 mJ when compared with that at 0%. When the $KHCO_3$ powder concentrations are 60% and 80%, the MIE values of PE reach 105 mJ and 175 mJ, respectively. For $K_2C_2O_4$, according to linearly fitted lines 1 and 2, it can be discovered that their corresponding slopes are 0.99 and 2.62, respectively. The critical concentration of $K_2C_2O_4$ powder is 40%, at which point the MIE rises by 39 mJ when compared with that at 0%. When the $K_2C_2O_4$ powder concentrations are 60% and 80%, the MIE values of PE reach 87 mJ and 154 mJ respectively, which are lower than that of the $KHCO_3$ powder.

By comparing the effects of two potassium salt powders, it can be found that the suppression effect of $KHCO_3$ is better than that of $K_2C_2O_4$. Since they both have the same potassium ion, the results further indicate that the bicarbonate ion has a greater suppression effect than the oxalate ion.

Combining the conclusions of the two groups compared in the test above, it can be found that the inhibitory effect of the bicarbonate ion on the MIT and MIE of PE dusts is better than that of the oxalate ion when the inert powders have the same particle size distribution and the same metal cation.

### 3.3. Comparison between the Two Alkali Metal Ions

In order to compare and analyze the effects of alkali metal ions (sodium ion and potassium ion) on the MIT and MIE of PE dusts, the four inert powders were divided into two groups (with $NaHCO_3$ and $KHCO_3$ as one group and $Na_2C_2O_4$ and $K_2C_2O_4$ as the other). The comparison results are shown in Figures 9 and 10, respectively.

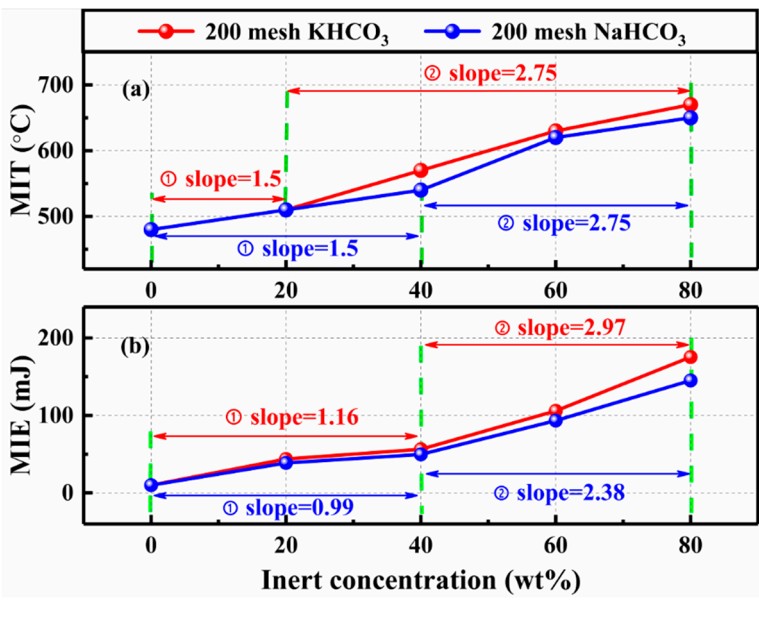

(a) MIT; (b) MIE

**Figure 9.** Effect of the $NaHCO_3$ and $KHCO_3$ powder concentrations on the MIT and MIE of PE.

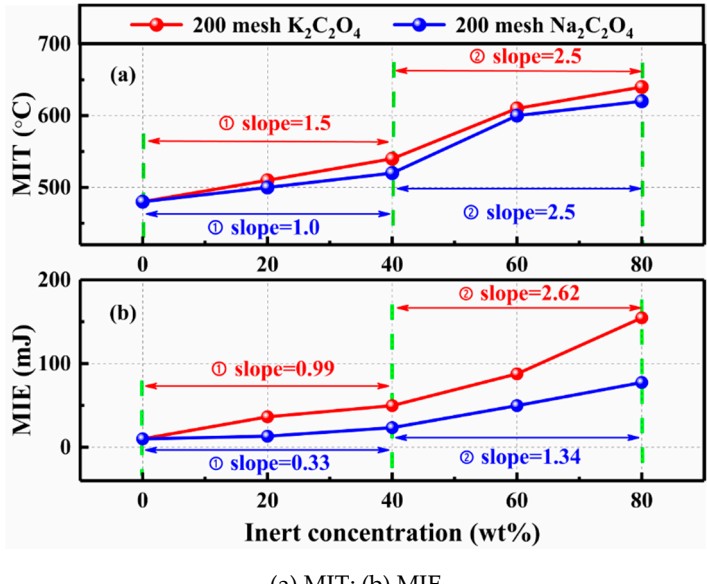

(a) MIT; (b) MIE

**Figure 10.** Effect of $Na_2C_2O_4$ and $K_2C_2O_4$ powder concentrations on the MIT and MIE of PE.

### 3.3.1. Effects of $NaHCO_3$ and $KHCO_3$ on the MIT and MIE of PE Dusts

The effects of $NaHCO_3$ and $KHCO_3$ on the MIT and MIE of PE dusts were compared, and the results are shown in Figure 9. In Figure 9a, it can be found that the suppression effect of $KHCO_3$ on the MIT of PE dust is better than that of $NaHCO_3$. According to the slopes of their linearly fitted lines, it is obvious that the critical concentration of $KHCO_3$ (20%) is lower than that of $NaHCO_3$ (40%). This indicates that $KHCO_3$ has a stronger suppression effect at lower concentrations than $NaHCO_3$. When both concentrations are increased to 80%, the MIT of PE dust suppressed by $KHCO_3$ increases to 670 °C, which is also higher than the 650 °C value for $NaHCO_3$. According to the effects of the two bicarbonate powders ($NaHCO_3$ and $KHCO_3$), it can be seen that the suppression effect of $KHCO_3$ on the MIT of PE dusts is better than that of $NaHCO_3$. In addition, it can be found from Figure 9b that the suppression effect of $KHCO_3$ on the MIE of PE is generally better than that of $NaHCO_3$. According to the slopes of their linearly fitted lines, it is obvious that although the critical concentration of $KHCO_3$ powder is similar to that of $NaHCO_3$ powder, the slope values of their linearly fitted number 1 lines are 1.16 and 0.99, respectively, indicating that a lower concentration $KHCO_3$ powder has a stronger suppression effect relative to $NaHCO_3$. When $KHCO_3$ concentrations are 60% and 80%, the MIE values of PE are 105 mJ and 175 mJ, respectively. When the $NaHCO_3$ powder concentrations are 60% and 80%, the MIE values of PE are 93 mJ and 145 mJ, respectively, which are lower than that of the $KHCO_3$ powder.

As mentioned above, it can be found that the suppression effect of $KHCO_3$ powder is better than that of $NaHCO_3$ powder on the MIT and MIE of PE dusts when the two bicarbonate powders have the same particle size distribution. In other words, the potassium ion has a greater suppression effect on the MIT of PE dusts than the sodium ion, because $NaHCO_3$ and $KHCO_3$ have the same bicarbonate ion.

### 3.3.2. Effects of $Na_2C_2O_4$ and $K_2C_2O_4$ on the MIT and MIE of PE Dusts

By comparing the effects of the two bicarbonate powders ($NaHCO_3$ and $KHCO_3$), it was discovered that the potassium ion has a greater suppression effect than the sodium ion on the MIT and MIE of PE dusts, because $NaHCO_3$ and $KHCO_3$ have the same bicarbonate ion. In order to further verify the conclusion above, the effects of two oxalate powders ($Na_2C_2O_4$ and $K_2C_2O_4$) on the MIT and MIE of PE dusts were tested and comparatively analyzed. The results are shown in Figure 10.

In Figure 10a, according to the two powders' first linearly fitted line, it is obvious that the slope of $K_2C_2O_4$ is larger than that of $Na_2C_2O_4$ and the values are 1.5 and 1.0, respectively. This indicates that $K_2C_2O_4$ has a stronger suppression effect at lower concentrations relative to $Na_2C_2O_4$, and the MIT values of PE dusts would rapidly increase when adding $K_2C_2O_4$. When the inert powder concentrations both are 40%, the MIT values under the action of $Na_2C_2O_4$ and $K_2C_2O_4$ are 520 °C and 540 °C, respectively. According to the slopes of linearly fitted line 2, it is obvious that although they had the same slopes of linearly fitted line 2, the MIT value suppression by $K_2C_2O_4$ is larger than that of $Na_2C_2O_4$. When their powder concentrations are 80%, the MIT suppression by $K_2C_2O_4$ also is higher than that of $Na_2C_2O_4$. Their values are 640 °C and 620 °C, respectively. According to the effects of the two oxalate powders ($Na_2C_2O_4$ and $K_2C_2O_4$), it can be seen that the suppression effect of $K_2C_2O_4$ on the MIT of PE dust is better than that of $Na_2C_2O_4$. Since $Na_2C_2O_4$ and $K_2C_2O_4$ have the same oxalate ion, it could be concluded that the potassium ion has a greater suppression effect on the MIT of PE dusts than the sodium ion. In addition, Figure 10b presents the effects of two oxalate powders ($Na_2C_2O_4$ and $K_2C_2O_4$) on the MIE of PE dusts. According to the two powders' linearly fitted lines, it is obvious that although their critical concentrations both are 40%, both slopes of $K_2C_2O_4$ are larger than those of $Na_2C_2O_4$. When their powder concentrations are 80%, the MIE of $K_2C_2O_4$ is higher than that of $Na_2C_2O_4$, where their values are 154 mJ and 77 mJ, respectively. As has been mentioned above, it could be concluded that the suppression effect of $K_2C_2O_4$ powder on both the MIT and MIE of PE dust is better than that of $Na_2C_2O_4$ powder when the two oxalate powders have same particle size distribution. In other words, the potassium ion has a greater suppression effect than the sodium ion on the MIT and MIE of PE dusts, because $Na_2C_2O_4$ and $K_2C_2O_4$ have the same oxalate ion.

According to the test and the comparative effects of the two groups (NaHCO$_3$ and KHCO$_3$, Na$_2$C$_2$O$_4$ and K$_2$C$_2$O$_4$), the results show that alkali metal ions (sodium ions and potassium ions) have a great suppression effect on the MIT and MIE of PE dusts. Meanwhile, potassium ions also present a better suppression effect than sodium ions.

### 3.4. Mechanism Analysis

The addition of an inert powder to combustible dust could reduce the ignition sensitivity of flammable dust. This effect could be attributed to several mechanisms, such as a cooling effect, a dilution effect, radiation absorption, turbulence modification, and the limitation of oxygen [24,25]. Meanwhile the heat balance of the fuel (inert mixture explosion addressed by Chatrathi and Going [26]) gives the idea that decomposition plays a key role in the inerting effectiveness, and this is consistent with Abbasi and Abbasi's research [27]. Hence, the thermal degradation behaviors of four inert powders (NaHCO$_3$, Na$_2$C$_2$O$_4$, KHCO$_3$, and K$_2$C$_2$O$_4$) were investigated by thermogravimetric (TG) and differential scanning calorimetry (DSC) methods under an air atmosphere, where the heating rate was 10 K/min. The TG and DSC curves were plotted, and they are shown in Figure 11. Meanwhile, the product ingredients of the PE dust combustion after adding four inert materials were analyzed by X-ray diffraction (XRD). The results are shown in Figure 12.

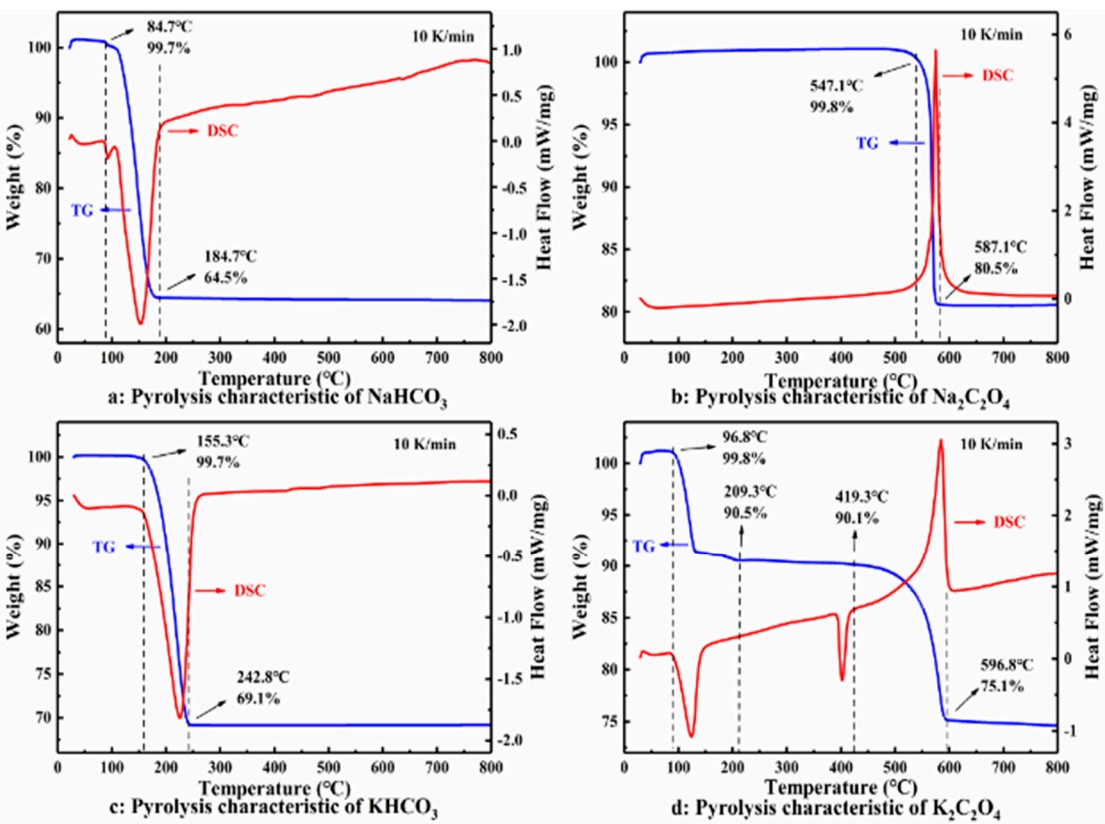

**Figure 11.** Pyrolysis characteristics of the four inert powders.

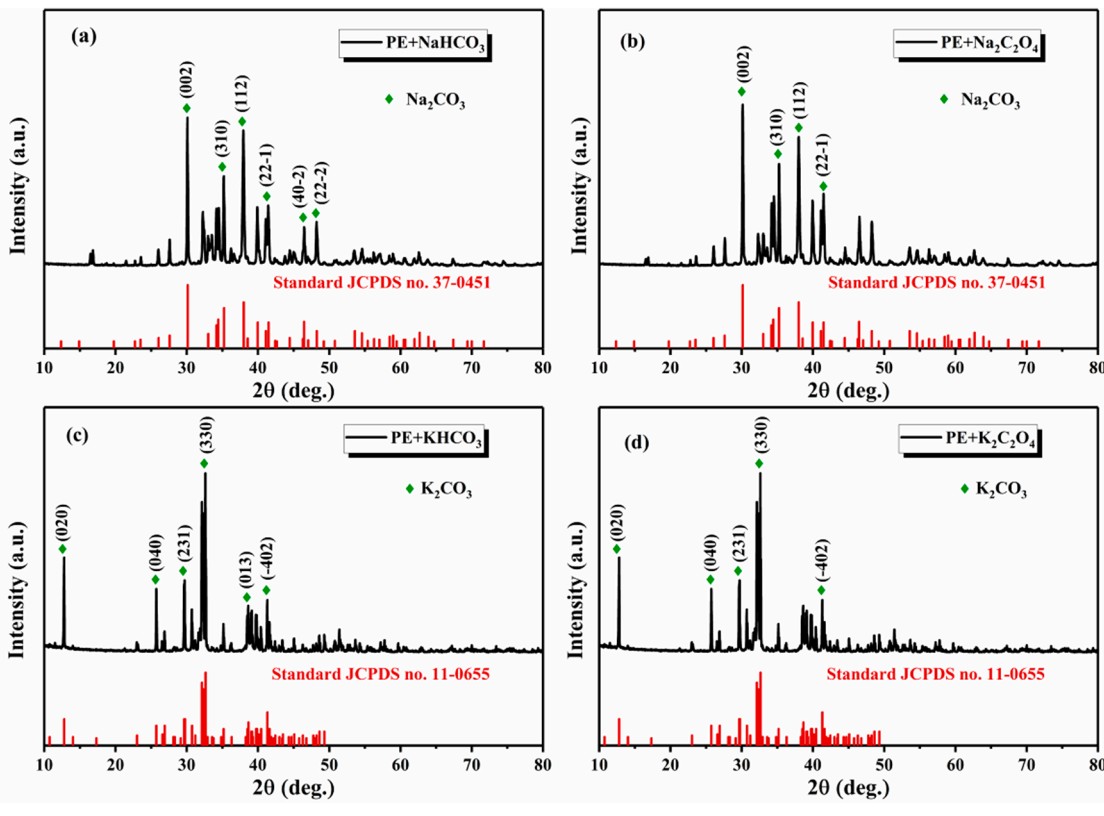

(a) PE + NaHCO₃; (b) PE + Na₂C₂O₄; (c) PE + KHCO₃; (d) PE + K₂C₂O₄

**Figure 12.** Product ingredient analyses of PE dust combustion after adding the four inert powders.

Figure 11a,c show the thermal degradation processes of the two bicarbonate salts powders (NaHCO₃ and KHCO₃). It was found that the NaHCO₃ and KHCO₃ powders both have only one stage of mass loss, which occurs in the temperature range of 84.7–184.7 °C and 155.3–242.8 °C, respectively. Moreover, it can be discovered that the NaHCO₃ and KHCO₃ powders both have an endothermic peak in the mass loss stage, with a heat absorption capacity of 522.6 J/g and 639.1 J/g, respectively. When the testing temperature is at around 184.7 °C, the thermal degradation processes of the NaHCO₃ powder are finished and the amount of residue is 64.5%. Then, when the testing temperature reaches around 242.8 °C, the thermal degradation processes of the KHCO₃ powder are finished, and the amount of residue is 69.1%. Meanwhile, Figure 12a,c show that the product ingredients of the PE dust combustion after adding two bicarbonate salts powders (NaHCO₃ and KHCO₃) are $Na_2CO_3$ and $K_2CO_3$, respectively. According to the thermal degradation processes of the NaHCO₃ and KHCO₃ powders and the product ingredient analyses, it can be observed that the two powders could decompose into $H_2O$, $CO_2$, and some carbonates ($Na_2CO_3$ and $K_2CO_3$) at a high temperature. This process can effectively absorb the heat from the experimental environment, resulting in an increase in the time and heat required to pyrolyze the PE particles to combustible gas. In addition, the decomposition products, i.e., $CO_2$ and $H_2O$, could shield heat conduction and thermal radiation and dilute the concentration of the combustible gases and oxygen in the reaction zone. Figure 11b,d show the thermal degradation processes of the two oxalate salts powders ($Na_2C_2O_4$ and $K_2C_2O_4$). It can be observed that the degradation onset temperatures of the $Na_2C_2O_4$ and $K_2C_2O_4$ powders are 547.1 and 96.8 °C, respectively. When the testing temperature is at around 587.1 °C, the thermal degradation processes of $Na_2C_2O_4$ powder are finished and the amount of residue is 80.5%. Then, when the testing temperature reaches around 596.8 °C, the thermal degradation processes of the $K_2C_2O_4$ powder are finished and the amount of residue is 75.1%. From the DSC curves of the $Na_2C_2O_4$ and $K_2C_2O_4$ powders, it can be found that the thermal decomposition of $Na_2C_2O_4$ and $K_2C_2O_4$ powder under heating generally

presents an exothermic process, with heat absorption capacities of $-435.8$ J/g and $-371.7$ J/g, respectively. Meanwhile, Figure 12b,d show that the product ingredients of PE dust combustion when adding two oxalate salt powders ($Na_2C_2O_4$ and $K_2C_2O_4$) are $Na_2CO_3$ and $K_2CO_3$, respectively. According to the thermal degradation processes of the $Na_2C_2O_4$ and $K_2C_2O_4$ powders and the product ingredient analyses, it can be observed that the two powders could decompose into $CO_2$ and some carbonates ($Na_2CO_3$ and $K_2CO_3$) at a high temperature. The inert gas that is generated can effectively decrease and dilute the concentration of the combustible gases and oxygen in the reaction zone. By comparing the thermal degradation behaviors of four inert powders ($NaHCO_3$, $Na_2C_2O_4$, $KHCO_3$ and $K_2C_2O_4$), it can be found that the heat absorption capacity of two bicarbonate salt powders ($NaHCO_3$ and $KHCO_3$) is better than that of two oxalate salt powders ($Na_2C_2O_4$ and $K_2C_2O_4$), and that the amount of residue of the two bicarbonate salt powders is lower than that of the two oxalate salt powders, indicating that the two bicarbonate salt powders can decompose into more inert products than the two oxalate salt powders under heating. Then, the two bicarbonate salt powders can present a greater suppression effect than the two oxalate salt powders on the ignition sensitivity of PE dusts.

Typically, the specific surface area of particles is related to their capacity to combine with free radicals. It is obvious that an inert powder with a larger specific surface area can effectively destroy and reduce free radicals and have a better inhibitory effect on dust combustion. In order to survey the surface adsorption properties, the four inert powders, with the same particle size, were investigated by the BET nitrogen adsorption method, and the results are shown in Table 1. According to the results of the specific surface area test, it can be seen that the specific surface area values of the four inert powders ($NaHCO_3$, $Na_2C_2O_4$, $KHCO_3$ and $K_2C_2O_4$) are 18.83, 1.419, 4.324, and 3.999 m$^2$/g, respectively. The specific surface area of the two bicarbonate salt powders ($NaHCO_3$ and $KHCO_3$) is larger than that of the two oxalate salt powders ($Na_2C_2O_4$ and $K_2C_2O_4$), indicating that $NaHCO_3$ and $KHCO_3$ powder can more effectively destroy and reduce the free radicals of PE dust, resulting in a better suppression effect on the ignition sensitivity of PE dust.

**Table 1.** Specific surface area of the four inert powders of same particle size.

| Inert Powder | Median Particle Diameter (μm) | Specific Surface Area (m$^2$/g) |
|---|---|---|
| $NaHCO_3$ | 77.5 | 18.83 |
| $Na_2C_2O_4$ | 77 | 1.419 |
| $KHCO_3$ | 75.1 | 4.324 |
| $K_2C_2O_4$ | 73.5 | 3.999 |

Regarding the chemical effects, the bicarbonate salts powders ($NaHCO_3$ and $KHCO_3$) can be pyrolyzed under heating to give some active groups, such as KOH and NaOH, which will chemically react with the highly reactive radicals (such as H and OH) generated during the ignition reaction of PE dusts. The availably of these can reduce the concentration of active free radicals in the system and further interrupt the chain reactions and suppress an explosion. Therefore, the reaction chain of combustion is interrupted, and the flame is extinguished. Overall, the bicarbonate salt powders mainly suppress dust combustion by relying on the chemical reaction of the decomposition products at a high temperature with the active free radicals generated by the PE dust combustion. The above process can be expressed by the following formulae [15,28]:

$$NaOH + H \Leftrightarrow Na + H_2O \tag{2}$$

$$NaOH + OH \Leftrightarrow NaO + H_2O \tag{3}$$

$$Na + OH + M \Leftrightarrow NaOH + M \tag{4}$$

$$KOH + H \Leftrightarrow K + H_2O \tag{5}$$

$$KO_2 + H \Leftrightarrow K + HO_2 \tag{6}$$

$$KO + OH \Leftrightarrow K + HO_2 \tag{7}$$

$$K + OH + M \Leftrightarrow KOH + M \tag{8}$$

Oxalate salt powders ($Na_2C_2O_4$ and $K_2C_2O_4$) will be pyrolyzed to produce some active groups (such as Na and K), which will chemically react with the highly reactive radicals (such as H and OH) generated in the PE dust combustion, resulting in a lower reaction temperature and reaction rate. The above process can be expressed by the following formulae [17,18]:

$$Na + OH \Leftrightarrow NaOH \tag{9}$$

$$NaOH + H \Leftrightarrow Na + H_2O \tag{10}$$

$$NaOH + OH \Leftrightarrow NaO + H_2O \tag{11}$$

$$NaO + H \Leftrightarrow NaOH \tag{12}$$

$$K + OH \Leftrightarrow KOH \tag{13}$$

$$KOH + H \Leftrightarrow K + H_2O \tag{14}$$

$$KOH + OH \Leftrightarrow KO + H_2O \tag{15}$$

$$KO + H \Leftrightarrow KOH \tag{16}$$

Under the dual action of the physical and chemical suppression effects, all four of the inert powders present suppression effects on the ignition sensitivity of PE dusts, resulting in an increase in the MIT and MIE.

## 4. Conclusions

In this study, the suppression effects of four inert powders ($NaHCO_3$, $Na_2C_2O_4$, $KHCO_3$, and $K_2C_2O_4$) on the MIT and MIE of PE dust have been investigated. According the experimental results, the suppression effects of the four inert powders have been analyzed and compared with each other. The conclusions of this work are summarized as follows:

All four inert powders of same particle size showed a significant suppression effect on the MIT and MIE of PE dust. Both the MIT and MIE increased with an increase in the inert powder concentration. The suppression effect of $KHCO_3$ powders is better than that of the other three powders. By comparing the suppression effects of three particle size distributions (80 mesh, 200 mesh, and 400 mesh) of $KHCO_3$ powder, it was concluded that $KHCO_3$ powder with a smaller particle size distribution has a better suppression effect.

By comparing the suppression effects of two groups of inert powders ($NaHCO_3$ and $Na_2C_2O_4$, $KHCO_3$ and $K_2C_2O_4$) on the MIT and MIE of PE dusts, the effects of two acid radical ions (bicarbonate ion and oxalate ion) were compared and analyzed. The results show that the two acid radical ions both present great suppression effects on the MIT and MIE of PE dusts, but the suppression effect of the bicarbonate ion was better than that of the oxalate ion when the inert powders were of the same particle size distribution and had the same metal cation.

By comparing the suppression effects of two groups (with $NaHCO_3$ and $KHCO_3$ as one group and $Na_2C_2O_4$ and $K_2C_2O_4$ as the other) on the MIT and MIE of PE dusts, the suppression effects of two alkali metal ions (sodium ion and potassium ion) were compared and analyzed. The results showed that the values of the MIT and MIE increased with the addition of sodium ions and potassium ions, meanwhile, the increasing values induced by potassium ions were greater than that by sodium ions, indicating that alkali metal ions (sodium ions and potassium ions) have a great suppression effect on both the MIT and MIE of PE dusts, and potassium ion have a better suppression effect than sodium ions.

Combining the physical and chemical effects, it was found that relative to oxalate salt powders, bicarbonate salt powders have better thermal degradation characteristics to effectively absorb heat from reaction zone, and they have a larger specific surface area. By comparing the heat absorption capacity of two groups of powders ($NaHCO_3$ and $KHCO_3$ as one group and $Na_2C_2O_4$ and $K_2C_2O_4$ as another), it can be found that potassium ions have a better heat absorption effect than sodium ions. Meanwhile, all four inert powders can be pyrolyzed to produce some active groups, which will chemically react with the highly reactive radicals generated in PE dust combustion. This results in a lower reaction temperature and reaction rate and has the effect of suppressing the MIT and MIE of PE dusts.

**Author Contributions:** C.L., Y.Q. and X.G. conceived and designed the experiments; W.J., C.L. and. performed the experiments and analyzed the data; Y.W. and X.W. managed all the experimental and writing process as the corresponding authors. All authors discussed the results and commented on the manuscript. All authors have read and agreed to the published version of the manuscript.

**Funding:** This research was funded by the National Natural Science Foundation of China (51874120, 51904094, 51504083, 51674104, 51806056), Basic Research Plan of Key Scientific Research Project of Henan University (20A620003); Program for Science & Technology Innovation Talents in Universities of Henan Province (19HASTIT042), the Research Foundation for Youth Scholars of Higher Education of Henan Province (2017GGJS053), the Fundamental Research Funds for the Universities of Henan Province (NSFRF1606), Program for Innovative Research Team in University of Ministry of Education of China (IRT_16R22), Program for Innovative Research Team of Henan Polytechnic University (T2018-2), Foundation for Distinguished Young Scientists of Henan Polytechnic University (J2017-3).

**Conflicts of Interest:** This research received no external funding.

## Acronyms

| | |
|---|---|
| MIE | The minimum ignition energy |
| MIT | The minimum ignition temperature |
| PE | Polyethylene |

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
