# Peer review of "Investigation into the Suppression Effects of Inert Powders on the Minimum Ignition Temperature and the Minimum Ignition Energy of Polyethylene Dust"

_processes, doi:10.3390/pr8030294_

Round 1
Reviewer 1 Report
Overall, the paper is well written and it includes sufficient explanations. However, within a scientific paper one should adopt a formal speech and use impersonal expressions. For example, at page 14, row 16, instead of "we need further studies", it should be "further studies are needed".
Within the text, you refer to the most of the figures as for example "7a" or "7b", but the figure itself does not show which is a and which is b. You should refer the couple of figures as the top and bottom charts or complete them with a and b letters.
Perhaps it is worth to include an Acronyms list at the beginning, where to explain MIE, MIT, PE, BET.
At page 23- up, "Specifically, the specific surface area (BET) of particles", maybe use "typically" instead of "specifically" in order to avoid the repetition.
Reviewer 2 Report
1-How temperature could change the potassium ion compared to sodium ion in the conclusion section?
2-How potassium ion could play a better significant impact on suppression than the sodium ion?It is needed to include the reason for that in the conclusion part.
